# Women Aged over 40 with Twin Pregnancies Have a Higher Risk of Adverse Obstetrical Outcomes

**DOI:** 10.3390/ijerph182413117

**Published:** 2021-12-12

**Authors:** María De la Calle, Jose L. Bartha, Laura García, Marcos J. Cuerva, David Ramiro-Cortijo

**Affiliations:** 1Department of Obstetrics and Gynecology, Hospital Universitario La Paz, 28046 Madrid, Spain; maria.delacalle@uam.es (M.D.l.C.); joseluisbartha@me.com (J.L.B.); marxichos@gmail.com (M.J.C.); 2Department of Pediatrics, Hospital Universitario Gregorio Marañón, 28007 Madrid, Spain; lauragiulia24@gmail.com; 3Department of Physiology, Faculty of Medicine, Universidad Autonoma de Madrid, 28029 Madrid, Spain

**Keywords:** twin pregnancy, aging, pregnancy complications, C-section, preterm labor

## Abstract

Maternal age is related to a higher risk of adverse maternal, fetal, and neonatal outcomes in twin pregnancies. However, whether this increase in adverse outcomes is due solely to age or due to risk factors that are more common in women over 40 remains unknown. The aim of this study is to assess if maternal age over 40 years old is an independent risk factor for obstetric adverse outcomes in dichorionic diamniotic twin gestations. In this single-center retrospective cohort study, we compared the obstetric outcomes of women with dichorionic diamniotic twin pregnancies below and over 40 years of age. A twin pregnancy cohort enrolled between 2013 and 2019 was included in the study. Maternal, fetal, and labor complications were recorded. A total of 510 women were analyzed in two groups: 266 women below 40 years old and 244 women over 40 years old. Maternal age over 40 increased the odds of maternal (aOR = 1.9 (1.3; 2.9); *p*-value = 0.002), fetal (aOR = 1.8 (1.0; 3.0); *p*-value = 0.037), and labor complications (aOR = 2.5 (1.3; 4.6); *p*-value = 0.004). Maternal age over 40 years was the most important factor increasing the odds of having a caesarean section (C-section). Over 40 years old was an independent risk factor for complications in dichorionic diamniotic twin pregnancies.

## 1. Introduction

The socioeconomic and lifestyle changes that have occurred mainly in developed countries over the last few decades have resulted in a delay of motherhood [1,2]. Several reasons have been suggested for this phenomenon such as the increase in women’s academic status, financial security and employment access, the family–work conflict, issues finding a suitable partner, the use of contraception, and family policies [3,4,5,6].

Health problems such as immune disease, hypertension, diabetes mellitus, or psychological difficulties, such as depression and anxiety, increase with age [7]. Therefore, advanced maternal age during pregnancy may have age-related physical and psychological complications during gestation. Furthermore, advanced maternal age is related to a higher rate of maternal, fetal, and obstetric complications [8,9,10]. In the FASTER trial, a multicenter study of singleton pregnancies, maternal age was found to be independently associated with specific adverse outcomes, such as miscarriage, congenital and chromosomal anomalies of the fetus, gestational diabetes, placenta previa, and a higher rate of C-sections [8]. Additionally, several observational studies indicate that advanced maternal age is an independent risk factor for preterm birth [11,12].

Unfortunately, many women are not fully aware of the effects delaying the first pregnancy has on the ability to conceive spontaneously, and many of them must resort to Assisted Reproductive Technology (ART). The risk of multiple pregnancies has been associated with these techniques [8,13]. Advanced maternal age is also related to a higher prevalence of twin pregnancies [14]. Therefore, the combined effect of the higher risk of spontaneously conceived twins associated with advanced maternal age and the use of ART accounts for the rise of twin pregnancies in women over 40 in most countries nowadays [15,16].

Twin pregnancy is considered a risk factor for preterm birth, low birth weight, fetal death, and adverse fetal outcomes [17]. In addition, women carrying twins are at a higher risk for maternal complications such as hypertension induced by pregnancy, preeclampsia, gestational diabetes, intrahepatic cholestasis, anemia, C-section, and thromboembolism [18,19,20,21,22].

In contrast to singleton pregnancies, current evidence in twin pregnancy and maternal age suggests that age itself does not affect pregnancy outcomes [10,23,24,25,26,27]. However, most of the studies considered women over 35 as advanced maternal age. So far, only two studies have evaluated twin pregnancies in women over 40 years, and both found a higher risk of preterm delivery [24,27]. Considering that the proportion of twin pregnancies in women over 40 years old is increasing, the aim of this study was to assess if advanced maternal age over 40 is an independent risk factor for adverse outcomes in dichorionic diamniotic twin pregnancies.

## 2. Materials and Methods

### 2.1. Study Design and Cohort Enrollment

This is a single-center retrospective, non-interventional, and observational cohort study. Pregnant women attending in the Obstetrics and Gynecology department of the Hospital Universitario La Paz (HULP, Madrid, Spain) between January 2013 and December 2019 were enrolled in this study.

The inclusion criteria were women with dichorionic diamniotic twin pregnancies who had obstetric follow-up and labor at HULP over 22 weeks of gestational age; HULP attended per year approximately 100 dichorionic diamniotic twin pregnancies. The exclusion criteria were maternal age below 18, non-dichorionic diamniotic twin pregnancies and incomplete medical data regarding to maternal age, use of ART, and adverse events during pregnancy or labor. A total of 622 twin pregnancies were attended at HULP during the study period, of which 510 matched the criteria and were finally included in the analysis.

This study was performed in accordance with the Declaration of Helsinki regarding studies in human subjects, and it was approved by the HULP Ethical Committees (PI-3560). Women were divided into two groups according to age at conception: below 40 (*n* = 266) and 40 years or older (over 40 years; *n* = 244). The flow chart to be analyzed in the study is shown in Figure 1.

### 2.2. Data Collection

Medical records, surgical protocols, and nursing documents were reviewed to record:

**Maternal variables at the beginning of pregnancy**: age (years), origin (European/non/European), parity, smoking habits (yes/no; considering women who smoked during pregnancy or those who had smoked less than 24 months prior to pregnancy as smokers), body mass index (BMI, Kg/m^2^), and chronic diseases (any disease during the previous 5 years, including obesity, hypertension, diabetes mellitus, cancer, immune diseases, hematologic diseases, rheumatologic diseases, and endocrinologic diseases).

**Reproduction variables**: type of conception (spontaneous/ART) and type of ART (ovarian stimulation, artificial insemination, in vitro fertilization (IVF), or oocyte donation).

**Maternal complications during pregnancy**: diagnosis of the following conditions at any time: hyperemesis, lower back pain, gastroesophageal reflux, thrombocytopenia (defined as platelets lower than 100 × 10^3^/mL), anemia (defined as hemoglobin lower than 11 g/dL), gestational diabetes mellitus (defined as a positive result in the 100 g oral glucose tolerance test), pregnancy-induced hypertension (defined as systolic blood pressure higher than 140 mmHg and/or diastolic blood pressure higher than 90 mmHg after 20 weeks of gestational age), preeclampsia (defined as systolic blood pressure higher than 140 mmHg and/or diastolic blood pressure higher than 90 mmHg and proteinuria over 300 mg in 24 h urine) and intrahepatic cholestasis (defined as fasting serum bile acid levels greater than 10 µm/L). Maternal complication variable was created as the presence of one of the any previous diagnoses, as previously published [28].

**Fetal complications during pregnancy**: diagnoses or events of fetal malformation of any fetus (including urogenital tract, circulatory system, nervous system, and umbilical cord), intrauterine growth restriction (IUGR; defined as fetal growth below third percentile or below tenth percentile with hemodynamic alterations), and fetal death of one or both fetuses. Fetal complication variable was created as the presence of one of the any previous diagnoses, as previously published [28].

**Obstetric complications during pregnancy**: diagnoses or event of threat of preterm labor, preterm premature rupture of membranes (defined as ocular assessment of amniotic fluid leakage from the cervix and/or reduced uterine amniotic fluid volume by ultrasound), preterm labor (labor before 37 weeks of gestation; also, it was categorized as <34 weeks, between 34 and 37 weeks, and ≥37 weeks), placenta previa, and placenta accrete. The obstetric complication during pregnancy variable was created as the presence of one of any of the previous diagnoses, as previously published [28].

**Labor variables**: gestational age (weeks), mode of delivery (vaginal/C-section), and birth weight (grams).

**Complications during labor**: informed event during labor of severe hemorrhage (loss of 1 L of blood within 24 h of delivery), hysterectomy, placental abruption, uterine rupture, and maternal death. Complication during labor variable was created as the presence of one of the any previous diagnoses, as previously published [28].

### 2.3. Statistical Analyses

Sample size was calculated after analyzing the maternal complications as the outcome of the calculation in twin dichorionic diamniotic pregnancies in our center during an entire year. We estimated that we would need 203 women in each group to demonstrate a difference of at least 15% in the rate of maternal complications with a confidence level of 0.95 and a statistical power of 80%. For practical reasons and considering that we follow around 100 dichorionic diamniotic twin pregnancies in our center every year, we decided to include all dichorionic diamniotic twin pregnancies followed between 2013 and 2019 in HULP.

Variable distribution was tested by the Kolmogorov–Smirnov test. Quantitative variables were expressed as mean ± standard deviation (SD) or median (Q1; Q3) as appropriate, and qualitative variables were expressed as relative frequencies (%) and sample size (*n*). Comparisons between groups were performed by Student’s *t*-test or Mann–Whitney’s U-test according to variable distribution. Proportion association was tested by Fisher’s exact test. The level of significance was set at 0.05. Binary logistic regression model was used to determine the factors that increased the odds of adverse maternal outcomes. The models were adjusted by significant maternal variables at the beginning of pregnancy. From the models, we extracted the adjusted odds ratio (aOR) and 95% confidence interval (CI). All analyses were performed using SPSS version 22.0 (SPSS Inc., Chicago, IL, USA).

## 3. Results

A total of 510 women were included in the study according to inclusion criteria (266 women under 40 and 244 over 40 years old). There were more European and nulliparous women and chronic disease rate in the over 40 than under 40 group. Contrary, the group of women under 40 had significantly more smokers than the other group. No differences in BMI were found between groups (Table 1).

More than half of all twin dichorionic diamniotic pregnancies were achieved by ART, being used in 47.7% in women below 40 and in 83.6% in women over 40 years of age. There was an association between ART and groups (*p*-value < 0.001). The rate of reproduction techniques was significantly different in both groups with a higher rate of insemination in the group below 40 and a more frequent use of in vitro fertilization (IVF) in the group of over 40 years. Nevertheless, the rates of ovary stimulation and oocyte donation were similar in both groups (Table 1).

Significant variables were used to adjust the binary regression models.

Overall, the prevalence of maternal complications was 63.9% (326) with anemia being the most common in both groups. However, there was a statistically significant association between groups and maternal complications, being more prevalent in women over 40 years (Table 2). Particularly, hyperemesis, anemia, gastroesophageal reflux, gestational diabetes, and preeclampsia rates were significantly higher in the women over 40 years. In addition, gestational diabetes was the only one significant complication different between women conceived spontaneously and ART (non-ART = 7.8%, ART = 17.0%; *p*-Value = 0.016).

Overall, fetal complications were 17.25% (88) of the dichorionic diamniotic twin pregnancies cohort. Fetal malformations were associated between groups, being more prevalent in women over 40 (Table 2). The most common malformations were single umbilical artery and fetal renal pelvis dilatation. Although without statistical significance, there were four cases of fetal death in the women below 40 and one in the women over 40. Three out of the five women who had a fetal death suffered chronic diseases (circulatory system disease, obesity, and asthma). Four of them were achieved by ART.

Gestational age at birth was 37.1 (36.0; 38.2) weeks for women below 40 and 37.1 (35.0; 37.9) weeks for women over 40. There was a significant difference between gestational age and groups (*p*-value = 0.001). However, when categorizing by gestational age, no significant differences in prematurity were detected (below 40: <34 weeks = 6.2%, 34–37 weeks = 29.5%, ≥37 weeks = 64.3%; Over 40: <34 weeks = 18.4%, 34–37 weeks = 25.5%, ≥37 weeks = 56.1%; *p*-value = 0.389). In addition, considering preterm delivery outcome, women over 40 years did not show statistical differences (1.0 (0.6; 1.5); *p*-value = 0.957). However, as expected, premature rupture of membrane was a risk factor for prematurity (13.5 (6.7; 31.3); *p*-value < 0.001). Overall, the rate of C-section was 67.2% (343). In the group of below 40 years, 52.6% (140) had C-sections, while 83.2% (203) in the group over 40 had C-sections (*p*-value < 0.001). C-section was significantly different between women who conceived spontaneously and ART (non-ART = 59.7%, ART = 77.7%; *p*-Value < 0.001).

Overall, the obstetric complications during pregnancy were 53.1% (271). The most frequent of them was preterm labor. Women over 40 years old presented significantly more obstetric complications during pregnancy than women below 40 years (Table 3). Particularly, there was a premature rupture of membrane and placenta accrete.

The complications during labor were present in 13.3% (68) of the cohort, with postpartum hemorrhage being the most common. Women over 40 years had significantly more complications during labor, such as severe hemorrhage and hysterectomy, than women below 40 years old (Table 3). There were no cases of uterine rupture or maternal death in our cohort.

After adjusting for confounders, the binary regression models showed that maternal age over 40 years significantly increased 1.9 (1.3; 2.9) times maternal complications, 1.8 (1.0; 3.0)-fold fetal complications and 2.5 (1.3; 4.6) times complications during labor (Table 4). Maternal age over 40 was also a factor that contributed to increasing the odds of C-section.

## 4. Discussion

Our results showed that being over 40 years old in dichorionic diamniotic twin pregnancies was an important factor that increases the fold of maternal, fetal, and labor complications during pregnancy. Maternal age greater than 40 is also an important risk factor for having a C-section in dichorionic diamniotic twin pregnancies.

Delayed childbearing is a complex phenomenon often found in modern societies. Women over 40 years of age have more fertility issues and diseases. Most of them conceived by ART and are at a higher risk of adverse outcomes [8,13,29]. Furthermore, these women are mostly nulliparous, which increases the rate of C-section and adverse events during labor.

Obstetricians and midwives following dichorionic diamniotic twin pregnancies should be aware of the effect that the combination of ART and advanced maternal age can have on the rate of maternal complications. Major complications such as preeclampsia and gestational diabetes increase in both twin pregnancies and advanced maternal age [8,11,18,19,20,21]. However, there are few reports regarding the combination of twin pregnancy and age above 40 years. Fox et al. reported that they did not find differences in the rates of preeclampsia and gestational diabetes in women over 40 carrying twins, although they only included 32 twin pregnant women over 40 in their study [25]. Preeclampsia, gestational diabetes, and anemia were more common among dichorionic diamniotic twin pregnant women over 40 in our cohort. In addition, the prevalence of gestational diabetes was higher in women conceived by ART than spontaneously. After adjusting for confounding factors, the most important factors that caused these conditions were advanced maternal age and the use of ART. In our opinion and supported by other authors, early supplementation with iron, low-dose aspirin, and vitamin D, healthy eating advice, and screening for preeclampsia and gestational diabetes should be considered since the first visit of the pregnancy to potentially avoid part of these complications [30,31].

Regarding complications during labor, the main factors increasing the odds in our cohort were maternal age greater than 40 and nulliparity. Women over 40 were at a higher risk of hemorrhage in our study. Previous studies did not report an increase in labor complications, such as hemorrhage, in twin pregnancies in women over 40 [23,24,25,26,27,32]. We did not consider C-section as a complication during labor, as most of these techniques were planned according to maternal preference. C-section was higher among women over 40, as previously reported by other authors [23,25,32]. Furthermore, our data showed that there were more C-sections in the group of ART women than in the group of gestations conceived spontaneously. The odds of C-section are a result of many interacting factors. The influence of culture, lifestyle, and environment on C-section has been discussed by several authors [33,34]. In our cohort, the factors involved in the odds of C-section were advanced maternal age, ART, nulliparity, smoking habits, and non-European origin.

There is a paucity of data on whether maternal age affects fetal complications such as stillbirth. Most reports regarding fetal death were inconclusive. Some authors have found an increase in the rate of stillbirth [27], while others reported a decrease in fetal deaths in twin pregnancies at high maternal ages [24,26]. Similarly, Lisonkova et al. reported an increase in the rate of small for gestational age and IUGR in women over 35 years old [26], while Delbaere et al. reported a lower risk in women over 35 [32]. Our results do not show an increase in fetal complications such as stillbirth or small for gestational age in the group of women over 40. We did encounter a higher rate of fetal malformations, although there were minor anomalies. There were more cases of intrauterine fetal death in the women below 40 years, although the results were not significant. We believe that the number of fetal ultrasound scans during pregnancy must be the same in dichorionic diamniotic twin pregnancies independently of maternal age.

Obstetrical complications during pregnancy, such as preterm labor, premature rupture of membranes, or placenta accrete were more closely related to factors such as ART or parity rather than maternal age over 40. Other authors have found similar results [23,32]. Nevertheless, McLennan et al. found an increase in preterm births [24], and Yang did in extremely preterm births with advanced maternal ages [27]. In our center, twin pregnancies, no matter the maternal age, are followed closely with monthly measurements of cervical length to predict and prevent prematurity. As a side note, the higher risk of postpartum hemorrhage and obstetric hysterectomy in the group of women over 40 years old could be related to the higher rate of placenta accrete found. In many cases, ART can be accompanied by metroplasty, which increases the incidence of placenta accrete.

### Limitations and Strengths

The main weakness of this study could be that both groups (below 40 and over 40 years old) were heterogenous in factors such as parity, ART, chronic diseases, or tobacco use. However, binary logistic regression models allowed assessing the impact of maternal age on the obstetric outcomes. Another limitation would be that we have only studied dichorionic diamniotic twins to avoid bias due to the specific complications that could arise for monochorionic twin pregnancies. Therefore, these results may not be applicable to other types of twin pregnancies.

The strength of the study would be the high number of dichorionic diamniotic twin pregnancies studied. This cohort study represents today’s real society in a western country and allows counseling for both parents and obstetricians before and during the pregnancy.

## 5. Conclusions

Women over 40 years of age who plan their first pregnancy through ART should be informed that they are exposed to a greater risk of maternal, obstetric, and fetal complications. According to our results, being over 40 years of age is an independent risk factor for maternal–fetal complications and complications during labor in dichorionic diamniotic twin pregnancies.

## Figures and Tables

**Figure 1 ijerph-18-13117-f001:**
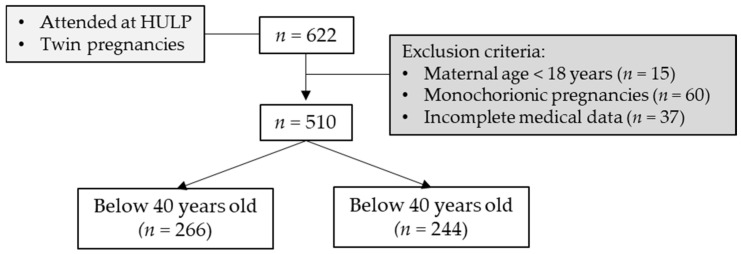
Flow chart of study participants. Sample size (*n*) is shown between brackets. HULP, Hospital Universitario La Paz.

**Table 1 ijerph-18-13117-t001:** Maternal characteristics and assisted reproduction techniques.

	Below 40 (*n* = 266)	Over 40 (*n* = 244)	*p*-Value
Maternal age (years)	34.0 (31.0; 34.0)	42.0 (41.0; 44.0)	<0.001
Body mass index (kg/m^2^)	23.0 ± 3.95	23.6 ± 3.51	0.098
European origin	94.0% (250)	97.9% (239)	0.024
Nulliparity	35.7% (95)	68.9% (168)	<0.001
Smoking habits	20.3% (54)	7.4% (18)	<0.001
Chronic diseases	25.9% (69)	35.7% (87)	0.017
Assisted reproduction techniques	47.7% (127)	83.6% (204)	<0.001
Ovary stimulation	0.3% (1)	0.0% (0)	1.0
Artificial insemination	4.9% (13)	1.2% (3)	0.021
In vitro fertilization	38.7% (103)	75.8% (185)	<0.001
Oocyte donation	3.8% (10)	6.6% (16)	0.151

In quantitative variables, the data were expressed as media ± SD or median (Q1; Q3) and the *p*-value was extracted of Student’s *t*-test or Mann–Whitney’s U-test according to the variable distribution. In qualitative variables, the data were shown as relative frequency and sample size (*n*) between brackets and the *p*-value was extracted of Fisher’s test.

**Table 2 ijerph-18-13117-t002:** Maternal and fetal complications during pregnancy.

	Below 40 (*n* = 266)	Over 40 (*n* = 244)	*p*-Value
**Maternal complications**	53.8% (143)	75.0% (183)	<0.001
Hyperemesis	12.4% (33)	20.9% (51)	0.010
Thrombocytopenia	5.3% (14)	4.5% (11)	0.693
Anemia	30.1% (80)	46.3% (113)	<0.001
Lower back pain	5.3% (14)	9.4% (23)	0.070
Gastroesophageal reflux	5.3% (14)	18.9% (46)	<0.001
Gestational diabetes	10.1% (27)	16.4% (40)	0.037
Pregnancy-induced hypertension	1.9% (5)	3.7% (9)	0.280
Preeclampsia	7.1% (19)	15.6% (38)	0.003
Intrahepatic cholestasis	2.3% (6)	5.3% (13)	0.067
**Fetal complications**	12.0% (32)	22.9% (56)	0.001
Malformations	4.9% (13)	16.0% (39)	<0.001
Intrauterine growth restriction	6.4% (17)	9.8% (24)	0.153
Fetal death	1.5% (4)	0.4% (1)	0.375

Data were shown as relative frequency and sample size (*n*) between brackets and *p*-value was extracted of Fisher’s test.

**Table 3 ijerph-18-13117-t003:** Obstetrical complications during pregnancy and labor.

	Below 40 (*n* = 266)	Over 40 (*n* = 244)	*p*-Value
**Obstetric complications during pregnancy**	44.7% (119)	62.3% (152)	<0.001
Threat of preterm labor	7.9% (21)	7.8% (19)	0.964
Premature rupture of membranes	9.8% (26)	16.4% (40)	0.026
Preterm labor	35.7% (95)	43.9% (107)	0.060
Placenta previa	0.4% (1)	2.5% (6)	0.058
Placenta accrete	0.4% (1)	3.3% (8)	0.016
**Complications during labor**	6.8% (18)	20.5% (50)	<0.001
Hemorrhage	6.8% (18)	19.3% (47)	<0.001
Hysterectomy	0.4% (1)	2.9% (7)	0.031
Placental abruption	0.0% (0)	1.2% (3)	0.109
Uterine rupture	0.0% (0)	0.0% (0)	-
Maternal death	0.0% (0)	0.0% (0)	-

Data were shown as relative frequency and sample size (*n*) between brackets, and the *p*-value was extracted from Fisher’s test.

**Table 4 ijerph-18-13117-t004:** Binary logistic regression models adjustment for confounders variables.

	aOR	95% CI	*p*-Value		aOR	95% CI	*p*-Value
**Maternal Complications**				**Fetal complications**			
Over 40 years	1.9	1.3–2.9	0.002	Over 40 years	1.8	1.0–3.0	0.037
ART	1.9	1.2–2.9	0.003	ART	3.8	1.9–7.5	<0.001
Nulliparous	1.2	0.8–1.8	0.390	Nulliparous	0.8	0.4–1.3	0.288
Smoker	0.9	0.5–1.5	0.663	Smoker	1.4	0.7–2.9	0.363
Maternal chronic disease	1.2	0.8–1.8	0.442	Maternal chronic disease	1.1	0.7–1.8	0.714
Non-European	1.2	0.5–3.0	0.733	Non-European	2.7	0.8–8.4	0.093
**Obstetric complications during pregnancy**				**Complications during labor**			
Over 40 years	1.3	0.9–2.0	0.179	Over 40 years	2.5	1.3–4.6	0.004
ART	1.6	1.1–2.5	0.022	ART	1.2	0.6–2.5	0.612
Nulliparous	2.2	1.5–3.4	<0.001	Nulliparous	2.4	1.3–4.7	0.008
Smoker	0.8	0.4–1.3	0.344	Smoker	0.5	0.2–1.6	0.256
Maternal chronic disease	1.3	0.9–2.0	0.155	Maternal chronic disease	1.3	0.8–2.3	0.301
Non-European	2.2	0.9–5.6	0.090	Non-European	1.5	0.3–7.2	0.615
**C-section**							
Over 40 years	3.2	2.0–5.0	<0.001				
ART	1.9	1.2–3.1	0.004				
Nulliparous	2.7	1.7–4.2	<0.001				
Smoker	1.9	1.0–3.4	0.038				
Maternal chronic disease	1.1	0.7–1.8	0.585				
Non-European	3.4	1.2–9.7	0.020				

Data shown adjusted odds ratio (aOR), 95% confidence intervals (CI) and *p*-values associated. The groups with women below 40 years were considered as a reference.

## Data Availability

The data presented in this study are available on request from the corresponding author. The availability of the data is restricted to investigators based in academic institutions.

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
