# Peer review of "Women Aged over 40 with Twin Pregnancies Have a Higher Risk of Adverse Obstetrical Outcomes"

_ijerph, 2021, doi:10.3390/ijerph182413117_

Round 1

Reviewer 1 Report

De la Calle et al has evaluated if advanced maternal age over 40 years is an independent risk factor for adverse pregnancy outcomes in dichorionic diamniotic twin pregnancy. This a retrospective cohort study including 510 women. They showed advanced maternal age per 40 years is associated with adverse pregnancy outcomes (after 22 weeks of gestation) including maternal, fetal and labor complications in dichorionic diamniotic twin gestations.

This is a nicely conducted study with very important clinical implications. I want to congratulate the authors.

After my review, I would like to share my comments and suggestions.

1-Introduction is brief and relevant.

2-Material and Method section is clear. I want to ask the authors to define preterm premature rupture of membranes. It will be very helpful if preterm delivery (delivery before 37 weeks) is included as a separate outcome. I also suggest that it will be very informative if the authors can separate preterm delivery as two subgroups such as before 34 weeks and before 37 weeks.

3-Results section is also clear with nicely put tables.

3a-I want to ask if and how the authors included preterm delivery as an outcome. There are two different outcomes included as “preterm labor” (PTL) and “preterm premature rupture of membranes” (PPROM). Did both of these outcomes end up with preterm delivery? That’s what I am assuming since there was threatened preterm labor as a different outcome category. I want to ask if authors can look into “preterm delivery” outcome as a separate outcome (including PPROM and preterm labor which ended with preterm delivery) since this is the clinically significant end point. Since PPROM comparison is statistically significant but PTL is not, I am wondering if overall preterm delivery outcome will be significantly different between two groups. The subgroup analysis of preterm delivery before 37 weeks and before 34 weeks will be clinically very helpful. I suggest that authors share this data as well and clearly explain this clinically important point in the text.

3b-Since ART especially IVF is used in majority of women over 40 years (more than 75-80% of their group), I want to ask authors if they can comment on if ART can have any effect on the statistically significant individual adverse outcomes (not as composite outcome) found in their analysis. The authors briefly mentioned this in their discussion section as after adjusting for confounding factors only maternal age and ART were found to be associated with the preeclampsia, GDM and anemia. There is supporting literature showing ART/IVF treatment may increase the adverse outcomes studied in this study. It is very important that authors clearly explain and emphasize this more for the readers. Thanks for including in the limitations section.

4-Discussion is very brief and clear. There is not enough data on this topic. Authors should clarify the suggestion on 3b here. I agree that these findings will help to plan a better screening and management in women over 40 years with dichorionic twin gestation.

Thank you

Author Response

De la Calle et al has evaluated if advanced maternal age over 40 years is an independent risk factor for adverse pregnancy outcomes in dichorionic diamniotic twin pregnancy. This a retrospective cohort study including 510 women. They showed advanced maternal age per 40 years is associated with adverse pregnancy outcomes (after 22 weeks of gestation) including maternal, fetal and labor complications in dichorionic diamniotic twin gestations.

This is a nicely conducted study with very important clinical implications. I want to congratulate the authors.

Response: Thank you very much for your time dedicated to our work and your kind words. Thank you for your comments, they have been of great help to improve our article considerably.

After my review, I would like to share my comments and suggestions.

  1. Introduction is brief and relevant.

Response: Thank you for your consideration.

  1. Material and Method section is clear. I want to ask the authors to define preterm premature rupture of membranes. It will be very helpful if preterm delivery (delivery before 37 weeks) is included as a separate outcome. I also suggest that it will be very informative if the authors can separate preterm delivery as two subgroups such as before 34 weeks and before 37 weeks.

Response: The definition of premature rupture of the membrane was included in the text. On the other hand, the categorization proposed by the reviewer was included, also as results in the text.

  1. Results section is also clear with nicely put tables.
  • I want to ask if and how the authors included preterm delivery as an outcome. There are two different outcomes included as “preterm labor” (PTL) and “preterm premature rupture of membranes” (PPROM). Did both of these outcomes end up with preterm delivery? That’s what I am assuming since there was threatened preterm labor as a different outcome category. I want to ask if authors can look into “preterm delivery” outcome as a separate outcome (including PPROM and preterm labor which ended with preterm delivery) since this is the clinically significant end point. Since PPROM comparison is statistically significant but PTL is not, I am wondering if overall preterm delivery outcome will be significantly different between two groups. The subgroup analysis of preterm delivery before 37 weeks and before 34 weeks will be clinically very helpful. I suggest that authors share this data as well and clearly explain this clinically important point in the text.

Response: This is a great point of discussion. Clinically, premature rupture of membranes (PROMs) is a risk factor for preterm delivery (PT). However, not all PROMs end in PT, and PT can be triggered without prior PROM. For this reason, they were considered independent events. Clinical advances can screen those women with PROMs and prevent PT. The definition and diagnosis of PROMs was added in the material and methods. Furthermore, PT showed no significant differences between maternal age groups (included in the results). However, the reviewer's interest has led us to include these data in the text. The PROM was a risk factor to lead PT in our cohort.

  • Since ART especially IVF is used in majority of women over 40 years (more than 75-80% of their group), I want to ask authors if they can comment on if ART can have any effect on the statistically significant individual adverse outcomes (not as composite outcome) found in their analysis. The authors briefly mentioned this in their discussion section as after adjusting for confounding factors only maternal age and ART were found to be associated with the preeclampsia, GDM and anemia. There is supporting literature showing ART/IVF treatment may increase the adverse outcomes studied in this study. It is very important that authors clearly explain and emphasize this more for the readers. Thanks for including in the limitations section.

Response: Thank you for this appreciation. After performing association analyses of individual complications and ART, only gestational diabetes and C-section was significantly different considering typing of women reproductive. These data were added in the text.

  1. Discussion is very brief and clear. There is not enough data on this topic. Authors should clarify the suggestion on 3b here. I agree that these findings will help to plan a better screening and management in women over 40 years with dichorionic twin gestation.

Thank you.

Response: Thank you for these kind words and the time spent on our work. The indications in section 3b were expanded in the discussion.

Reviewer 2 Report

This is a retrospective observational study on a large cohort of dichorionic diamniotic (DCDA) twins in women below and over 40 years of age. 

The aim of the study was to explore whether maternal age over 40 is an independent factor for adverse pregnancy outcomes in women over 40. In the study period (2013-2019) in the unit there were 622 twin pregnancies with 510 DCDA twins. The authors used suitable statistic comparison in between groups and logistic regression to correct for confounding factors and demonstrated that advance maternal age over 40 is an independent risk factor for adverse pregnancy outcomes in DCDA twin pregnancies in women over 40.

I agree that this paper should be published. 

I however have a few questions: 

Would it be possible to add a flowchart and to explain whether the pregnancies that were excluded where monochorionic?    

Why 22 weeks for viability? not 24?

What about the date on ovocite donation in the > 40 group. It is understandable if you do not have that data, however I would suspect that fetal complications would be higher with own ovocites in this group.

Also - the composite outcome - some of the pregnancy related complications are not so severe as to be included as such: gastroesophageal reflux, back pain. 

I have read the power calculation in the Method section, it was not clear to me what was the outcome you used for that.

Minor corrections and typos:

Title – suggestion: Women aged over 40 with twin pregnancies have a higher risk of adverse obstetrical outcomes

Abstract

Maternal age is related to a higher risk of adverse maternal, fetal and neonatal outcomes in twin pregnancies.

Line 18 was

C- section – caesarean section (C-section) line 23

Line 37 of maternal

Line 67 attending

Line 71 – why paranthesis?

Line 165 significance

Line 168 were achieved by ART

Table 2 hiperemesis

Author Response

This is a retrospective observational study on a large cohort of dichorionic diamniotic (DCDA) twins in women below and over 40 years of age.

The aim of the study was to explore whether maternal age over 40 is an independent factor for adverse pregnancy outcomes in women over 40. In the study period (2013-2019) in the unit there were 622 twin pregnancies with 510 DCDA twins. The authors used suitable statistic comparison in between groups and logistic regression to correct for confounding factors and demonstrated that advance maternal age over 40 is an independent risk factor for adverse pregnancy outcomes in DCDA twin pregnancies in women over 40.

I agree that this paper should be published.

Response: Thank you very much for your time dedicated to our work and your kind words. Thank you for your comments, they have been of great help to improve our article considerably.

I however have a few questions:

  • Would it be possible to add a flowchart and to explain whether the pregnancies that were excluded where monochorionic?

Response: The flow chart was added in the text.

  • Why 22 weeks for viability? not 24?

Response: In this study we did not consider fetal viability as a discriminating variable in the results.

  • What about the date on ovocite donation in the > 40 group. It is understandable if you do not have that data, however I would suspect that fetal complications would be higher with own ovocites in this group.

Response: This is a very good comment. Unfortunately, we did not have the data on whether the ovular donation was her own or not. However, we wanted to explore this aspect in more detail. Performing the analyses to compare obstetric comorbidities between oocyte donation (n=16) and the rest of ART only in women over 40 years of age. However, we did not reach statistical significance in any of the cases. The highest prevalence was found in anemia, with both groups presenting rates of 50% (p-value=0.998).

  • Also - the composite outcome - some of the pregnancy related complications are not so severe as to be included as such: gastroesophageal reflux, back pain.

Response: We strongly agree with this comment. However, we wanted to include these complications as they tend to be common in pregnancies. Furthermore, the gastroesophageal reflux and back pain in twin pregnancy are more frequent and severe than in single pregnancy and are considered complications rather than physiological gravity modifications. In addition, previous data from our group demonstrated that statistical power was achieved when obstetric complications are considered together (PMID: 27780538).

  • I have read the power calculation in the Method section; it was not clear to me what was the outcome you used for that.

Response: For the calculation of the sample size, we considered the summary variable of maternal complications as the outcome. We have updated this information in the text. Considering that between both groups of women (below 40 and over 40) there would be a difference in the ratio of maternal complications, at least 15%, as a clinically relevant.

Minor corrections and typos:

  • Title – suggestion: Women aged over 40 with twin pregnancies have a higher risk of adverse obstetrical outcomes

Response: We have considered your comments and we have modified the title.

  • Abstract - Maternal age is related to a higher risk of adverse maternal, fetal and neonatal outcomes in twin pregnancies.

Response: We have modified the line.

  • Line 18 was C- section – caesarean section (C-section) line 23 / Line 37 of maternal / Line 67 attending

Response: Thank you for this appreciation that were corrected in the text.

  • Line 71 – why paranthesis?

Response: Parenthesis were removed.

  • Line 165 significance / Line 168 were achieved by ART / Table 2 hiperemesis

Response: Thank you for this appreciation that were corrected in the text.